# Live-Cell Analysis of Human Cytomegalovirus DNA Polymerase Holoenzyme Assembly by Resonance Energy Transfer Methods

**DOI:** 10.3390/microorganisms9050928

**Published:** 2021-04-26

**Authors:** Veronica Di Antonio, Giorgio Palù, Gualtiero Alvisi

**Affiliations:** Department of Molecular Medicine, University of Padua, 35122 Padova, Italy; veronicadiantonio@libero.it (V.D.A.); giorgio.palu@unipd.it (G.P.)

**Keywords:** DNA polymerase, BRET, qBRET, UL44, ppUL44, UL54, pUL54, protein-protein interaction, DNA binding, conformation

## Abstract

Human cytomegalovirus (HCMV) genome replication is a complex and still not completely understood process mediated by the highly coordinated interaction of host and viral products. Among the latter, six different proteins form the viral replication complex: a single-stranded DNA binding protein, a trimeric primase/helicase complex and a two subunit DNA polymerase holoenzyme, which in turn contains a catalytic subunit, pUL54, and a dimeric processivity factor ppUL44. Being absolutely required for viral replication and representing potential therapeutic targets, both the ppUL44–pUL54 interaction and ppUL44 homodimerization have been largely characterized from structural, functional and biochemical points of view. We applied fluorescence and bioluminescence resonance energy transfer (FRET and BRET) assays to investigate such processes in living cells. Both interactions occur with similar affinities and can take place both in the nucleus and in the cytoplasm. Importantly, single amino acid substitutions in different ppUL44 domains selectively affect its dimerization or ability to interact with pUL54. Intriguingly, substitutions preventing DNA binding of ppUL44 influence the BRET_max_ of protein–protein interactions, implying that binding to dsDNA induces conformational changes both in the ppUL44 homodimer and in the DNA polymerase holoenzyme. We also compared transiently and stably ppUL44-expressing cells in BRET inhibition assays. Transient expression of the BRET donor allowed inhibition of both ppUL44 dimerization and formation of the DNA polymerase holoenzyme, upon overexpression of FLAG-tagged ppUL44 as a competitor. Our approach could be useful both to monitor the dynamics of assembly of the HCMV DNA polymerase holoenzyme and for antiviral drug discovery.

## 1. Introduction

DNA replication in Herpesviridae members is both an important model for the study of eukaryotic DNA replication and the target of several drugs approved for antiviral treatment. The entire process takes place within the host cell nucleus [1] and is mediated by a number of cellular and viral proteins [2,3]. The latter include a DNA polymerase holoenzyme, composed by a DNA-dependent DNA polymerase catalytic subunit (pol) and a DNA polymerase accessory protein (PAP) conferring processivity to pol. Such herpesviral DNA polymerase holoenzymes present several similarities and important differences. The crystal structures of the PAPs from several Herpesviridae members, including pUL42 from the α-*herpesvirinae* Herpes simplex 1 (HSV-1), ppUL44 from the β-*herpesvirinae* human cytomegalovirus (HCMV), as well as BMRF1 and PF-8 from the γ-*herpesvirinae* Epstein–Barr virus and Kaposi-associated herpesvirus (KHSV), respectively, have been solved [4,5,6,7]. Despite the low homology, all proteins display a remarkably similar fold, reminiscent of that of PCNA [8]. Likewise, they can all be topologically divided in two domains separated by a connector loop that is responsible for binding to pol [9], and they all present a basic face capable of mediating DNA binding through electrostatic interactions [10,11]. However, while PCNA is a trimer, and therefore needs clamp loaders and ATP for DNA binding, pUL42 is a monomer, while both ppUL44 and PF-8 form head-to-head dimers. This evidence explains the ability of herpesviral PAPs to bind to dsDNA in the absence of clamp loaders and ATP [12]. Among them, only ppUL44 possess the so-called “gap loop”, a highly flexible basic loop located in the N-terminal domain, which is not visible in the crystal structure, and further contributes to dsDNA binding [4]. Additionally, in all known cases, pol binds to the connector loop of its respective PAP via its C-terminal amino acids [9,13,14].

HCMV ppUL44 is a 52-KDa protein essential for viral replication, which binds to dsDNA and directly interacts with pUL54, stimulating its activity by tethering the DNA polymerase holoenzyme to the DNA template [12,15,16,17]. ppUL44 can be functionally and structurally divided into an N-terminal (residues 1–290) and a C-terminal domain (residues 291–433; see Figure 1). The N-terminal domain has been successfully crystallized and retains all known ppUL44 biochemical properties [4,12]. However, the C-terminal domain, responsible for ppUL44 transactivation properties and its phosphorylation-dependent nuclear transport, is absolutely required for virus replication [18,19,20,21,22]. UL44(1–290) adopts a C clamp-shaped structure and forms head-to-head dimers [9,23]. Dimerization involves interaction of six main-chain-to-main-chain hydrogen bonds and extensive packaging of hydrophobic side chain at the interface and results in the formation of a central cavity, able to accommodate the viral DNA [24]. Indeed, the ppUL44-dsDNA interaction depends on electrostatic interactions between the dsDNA backbone and basic residues located both on the central cavity and the gap loop [24]. Accordingly, the substitution of specific hydrophobic residues at the homodimerization interface of UL44(1–290) is sufficient to impair dimerization, and it also reduces the DNA binding affinity of ppUL44 in vitro [4]. Recent studies showed that dsDNA binding of ppUL44 is essential for HCMV DNA replication, as in the case of HSV-1 [25]. Indeed, substitution of either basic gap loop residues, or of residues at the dimerization interface such as L86 and L87 [4], cause a dramatic alteration of ppUL44 subcellular localization and reduce DNA binding in cells, thus completely abolishing the ability of ppUL44 to trans-complement oriLyt-dependent DNA replication [13,23,26]. Therefore, ppUL44 dimerization is important to stabilize the interaction between the pUL54 catalytic subunit and DNA, and an alteration of this dimerization may block DNA synthesis and HCMV replication [23,26]. In this context, both ppUL44 interaction with itself and with pUL54 emerged as potential targets for the development of novel antiviral approaches [27,28,29].

In this study we developed quantitative assays to monitor both processes in living cells, taking advantage of fluorescence and bioluminescence resonance energy transfer (FRET and BRET) methods. We applied such methods to a series of ppUL44 substitution derivatives selectively impaired in key properties such as the ability to bind to either dsDNA or pUL54, to self-interact, as well as being translocated to the nucleus. In particular, while FRET assays allowed to visualize protein subcellular localization, BRET saturation experiments provided the possibility to estimate the affinity of each interaction and detect hypothetic conformational changes of protein complexes using the B_50_ and B_max_ values, respectively.

Our results further confirm that both ppUL44 dimerization and its interaction with pUL54 rely on different key residues of ppUL44. Both events can take place either in the nucleus or in the cytosol, and occur in cells with similar affinity. We also developed a number of cell lines stably expressing several ppUL44 and pUL54 -YFP and -Renilla luciferase (RLuc) fusion proteins, and combinations thereof, to compare the utility of different expression strategies in BRET inhibition assays, upon expression of the BRET pairs in the presence of excess concentrations of competitors. Our results indicate that, while stable expression of both RLuc and YFP fusions results in suboptimal inhibition of protein interactions, transient expression of RLuc fusions in the presence of FLAG-tagged ppUL44, but not of its HSV-1 homologue pUL42, leads to concentration-dependent inhibition of both ppUL44 dimerization and its interaction with pUL54.

## 2. Materials and Methods

### 2.1. Plasmids 

Mammalian expression plasmids were generated using the Gateway^TM^ technology (Thermofisher Scientific, Waltham, MA, USA). Entry clones pDNR-UL44(2–433); pDNR-UL44(2–433)L86A/L87A encoding for a dimerization defective protein; pDNR-UL44(2–433)I135A, impaired in binding to pUL54; pDNR-UL44(2–433)Δloop, bearing the R165G/K167N/K168G triple substitution in the DNA–binding flexible gap loop; pDNR-UL44(2–433)ΔNLS, bearing the K428V/K429A/K431 triple substitution within ppUL44-NLS; pDNR-UL44(405–433), encoding for ppUL44 C-terminal C2N motif, comprising the nuclear localization signal (NLS) and nuclear import-regulating phosphorylation sites; pDNR-UL54(1125–1242), encoding for pUL54 C-terminal residues encompassing the NLS and the ppUL44 binding site; pDNR-NLS[G], encoding a short peptide conferring slight nuclear accumulation to reporter proteins, were previously described [13,18,26,30]. A schematic representation of ppUL44 derivatives tested in this study is shown in Figure 1. Entry clones were used to generate C-terminal fusion mammalian expression vectors, following LR recombination reactions with the pDESTnYFP, pDESTnCFP, pDESTnRLuc Gateway compatible vectors [31], as described in [32]. Plasmid pWPI-puro-YFP-UL44 was generated by ligation a NheI-XbaI fragment from pDESTnYFP-UL44 in vector pWPI-puro. All constructs were confirmed by sequencing. Plasmids mediating expression of CFP-YFP, CFP and YFP used as positive or negative controls for FRET experiments were described in [33]. Plasmids pDESTnFLAG-UL44 and pDESTnFLAG-UL42 were described in [34]. Packaging plasmid pCMVΔ8.74 (#22036, Addgene, Watertown, MA, USA) and the VSV envelope glycoprotein expression vector pMD2.G (#12259, Addgene) were kindly provided by Didier Trono (Lausanne, Switzerland).

### 2.2. Cell Culture

HEK293T and HEK293A cells were maintained in Dulbecco’s modified Eagle’s medium (DMEM) supplemented with 10% (*v*/*v*) fetal bovine serum (FBS), 50 U/mL penicillin, 50 U/mL streptomycin and 2 mM L-glutamine and passaged when reached confluence.

### 2.3. Western Blotting Analysis

HEK293T cells were seeded in 6-well plates (5 × 10^5^ cells/well) and the next day were transfected with 5 μg of Renilla luciferase (RLuc)-expression constructs, using the calcium phosphate method as described in [35]. At 48 h post transfection, cells were washed with ice-cold PBS and lysed in 250 μL of cracking buffer (0.09 M Tris·Cl, pH 6.8. 20% glycerol, 2% SDS, 0.02% bromophenol blue, 0.1 M DTT). Ten microliters of each sample was analyzed by SDS page/Western blotting as described previously [26], using the α-RLuc mouse monoclonal antibody (Ab; MAB4400, Merck Millipore, Burlington, MA, USA; 1:4000) and the goat α-mouse immunoglobulin Ab conjugated to horseradish peroxidase (sc-2055, Santa Cruz Biotech, Dallas, TX, USA; 1:10,000), both diluted in PBS/BSA 3% (*w/v*). Immunoblots were developed with the ECL prime substrate (Amersham, Little Chalfont, UK) in combination with Carestream^®^ Kodak^®^ BioMax^®^ light autoradiography films (Merck Millipore).

### 2.4. Indirect Immunofluorescence Analysis

HEK293T cells were seeded in a 24-well plate onto glass coverslips (1 × 10^5^ cells/well) and the next day were transfected with 250 ng of RLuc-expression constructs, using Lipofectamine 2000 (Thermofisher Scientific), following the manufacturer’s recommendations. At 48 h post-transfection, cells were washed with PBS and fixed with 4% paraformaldehyde for 10 min at RT. Cells were permeabilized for 10 min at RT with PBS/Triton 0.1% (*v*/*v*). After blocking overnight at 4 °C in PBS/BSA 5% (*w*/*v*), cells were incubated 1 h at 37 °C in a humidified chamber with primary α-RLuc monoclonal Ab (MAB4400, Merck Millipore; 1:40, in PBS/BSA 1% (*w*/*v*)). Following 3 washes with PBS, cells were further incubated for 1 h at RT with Alexa 568 conjugated rabbit anti mouse secondary Ab (A10037, Thermofisher Scientific; 1:1000, in PBS/BSA 1% (*w*/*v*)). Following three washes with PBS1x and one with milliQ water, the coverslips were mounted on glass slides with Fluoromount G (Southern Biotech, Birmingham, Alabama, USA). Subcellular localization of fusion proteins was analyzed using a Leica SP2 confocal laser scanning microscope (Leica Microsystems, Wetzlar, Germany) equipped with a 63× oil immersion objective as described in [36].

### 2.5. FRET Acceptor Photobleaching Assays

The effects of specific substitutions on the ability of ppUL44 to self-interact or to bind the catalytic subunit pUL54 were assessed by confocal laser scanning microscopy (CLSM), as described above, and subjected FRET acceptor photobleaching, as described in [33]. Briefly, HEK293T cells were seeded in a 24-well plate onto glass coverslips (1 × 10^5^ cells/well) and the next day were transfected with YFP and CFP expression constructs (50–250 ng depending on the expression plasmid), using Lipofectamine 2000 (Thermofisher Scientific), following the manufacturer’s recommendations. At 48 h post transfection, cells were washed with PBS and fixed with 4% paraformaldehyde for 10 min at room temperature. After a wash with milliQ water, the samples were mounted on glass slides with Fluoromount G (Southern Biotech). FRET efficiency was determined according to the following formula: FRETeff = [(EDpost − EDpre)/EDpost] × 100, where ED represents the emitted donor fluorescence before (EDpre) or after (EDpost) photobleaching of the acceptor fluorophore. Only cells expressing similar levels of either CFP or YFP fusion proteins were analyzed (CFP and YFP intensity comprised between 50 and 200 arbitrary units). Measurements of at least 20 cells from two independent transfections were used to calculate the box plots. Images were processed using ImageJ (NIH).

### 2.6. BRET Assays

BRET experiments were performed essentially as described in [31,37]. HEK293T cells were seeded in a 24-well plate (1 × 10^5^ cells/well) and the next day were transfected using Lipofectamine 2000 (Thermofisher Scientific) following the manufacturer’s recommendations, with appropriate amounts of BRET donor and BRET acceptor expressing plasmids (5–500 ng depending on the expression plasmid). For each construct, the donor (RLuc)-expressing plasmid was transfected both in the absence and in the presence of the relative acceptor (YFP)-expressing plasmid to allow calculation of background BRET signal. At 48 h post transfection, culture medium was removed from wells, and cells were gently washed with 1 mL of PBS, before being resuspended in 290 µL of PBS. Cell resuspensions (90 µL) were then transferred to a 96-well black flat-bottom polystyrene TC-treated microplate (#3916, Corning, Corning, NY, USA) in triplicate, and signals were acquired using a reader compatible with BRET measurements (VICTOR X2 Multilabel Plate Reader, Perkin Elmer, Waltham, MA, USA). Fluorescence signals (YFPnet) relative to YFP fluorescence emission were acquired using a fluorometric excitation filter (band pass 485 ± 14 nm) and a fluorometric emission filter (band pass 535 ± 25 nm). Luminometric readings were performed at 5, 15, 30, 45 and 60 min after addition of native Coelenterazine (PJK Biotech, Kleinblittersdorf, Germany; 5 μM final concentration in PBS) as described in [38]. Data were acquired for 1 s/well, using a luminometric 535 ± 25 nm emission filter (YFP signal) and a luminometric 460 ± 25 nm emission filter (RLuc signal). Before reading, the plate was shaken for 1 s at normal speed and with double orbit. After background subtraction using values relative to mock transfected cells, the data obtained were used to calculate the BRET signal, defined as the ratio between the YFP and RLuc signals calculated for a specific BRET pair, according to the formula:BRET signal = (YFP emission)/(RLuc emission)

Similarly, the BRET ratio, defined as the difference between the BRET value relative to a BRET pair and the BRET value relative to the BRET donor alone, was calculated according to the formula:BRET ratio = (YFP emission)/(RLuc emission) BRET pair − (YFP emission)/(RLuc emission) BRET donor

For BRET saturation experiments, HEK293T cells were transfected with appropriate amounts of RLuc and YFP expression plasmids, ensuring similar levels of RLuc expression. BRET saturation curves were generated using the GraphPad Prism software (Graphpad Software Inc., San Diego, California, USA) by plotting each individual BRET ratio value to the YFPnet/RLuc signal, and interpolating such values using the one-site binding hyperbola function of GraphPad Prism (Graphpad Software Inc.) to calculate BRET_max_ (B_max_) and BRET_50_ (B_50_) values, indicative of maximum energy transfer and relative affinity of each BRET pair tested, respectively [39]. For BRET inhibition experiments, cells were transfected with appropriate BRET plasmids to generate a BRET signal similar to 50% of the BRET_max_ value, in the presence of increasing concentrations of either pDESTnFLAG-UL44 or pDESTnFLAG-UL42, and the BRET ratio values were plotted against the concentration of pDESTnFLAG plasmid used for transfection.

### 2.7. Generation of Lentiviral Particles

For production of lentiviral particles, 1.2 × 10^6^ HEK293T cells were seeded into 6 cm diameter dishes and transfected with the calcium phosphate method as described above, using 6.4 μg packaging plasmid (pCMVΔ8.91), 6.4 μg of pWPI-YFP-UL44 and 2.1 μg of the VSV envelope glycoprotein expression vector (pMD2.G). Forty-eight and 72 h post-transfection, supernatant containing lentiviral particles was harvested and filtered through a 0.45 μm pore membrane prior to usage for transducing HEK293A cells.

### 2.8. Generation of Polyclonal and Monoclonal HEK293A Cells Stably Expressing YFP-UL44

A total of 2 × 10^5^ HEK293A cells were seeded in 12-well plates and transduced three times every 12 h with 2 mL of cell culture supernatants containing YFP-UL44 encoding lentiviral particles, to achieve high number of integrates and, thus, high expression levels. Transduced cell pools were subjected to selection with medium containing puromycin (A1113803, Thermofisher Scientific; 1 μg/mL), as described in [40] to generate the polyclonal cell line HEK293A YFP-UL44 (thereafter called 68 × 3). To isolate monoclonal cell lines expressing different levels of YFP-UL44, 66 × 3 polyclonal cells were seeded in 96-well plates at a concentration of 0.5 cells/well and further cultured in the presence of 1 μg/mL puromycin (A1113803, Thermofisher Scientific; 1 μg/mL). Wells containing one single cell were further monitored daily for expression of fluorescence and expanded upon confluency. When surface area occupied by each cell clone reached 150 cm^2^, cell stocks were frozen in liquid nitrogen for further usage.

### 2.9. Generation of Polyclonal and Monoclonal HEK293A Cells Stably Expressing YFP-UL44 and Either Rluc-NLS[G] or Rluc-UL44

A total of 5 × 10^5^ monoclonal HEK293A YFP-UL44 expressing cells (clone 1B2) were seeded in 6-well plates and transfected with 10 μg of either pDESTnRLuc-UL44 or pDESTnRLuc-NLSG, with the calcium phosphate method as described above. Transduced cell pools were subjected to selection with medium containing geneticin (11811098, Thermofisher Scientific; 750 μg/mL) and puromycin (A1113803, Thermofisher Scientific; 1 μg/mL), as described in [35] to generate a polyclonal cell line 1B2/Rluc-UL44 and 1B2/Rluc-NLS[G]. To isolate monoclonal cell lines expressing different levels of Rluc fusion proteins, both polyclonal double cell lines were seeded in 96-well plates at a concentration of 0.5 cells/well and further cultured in the presence of 750 μg/mL gentamicin. Only wells containing one single cell/well were further considered, monitored daily for expression of fluorescence and expanded when confluency was reached, until surface area occupied by each cell clone reached 150 cm^2^, when cell stocks were frozen in liquid nitrogen for further usage.

### 2.10. Statistical Analysis

Statistical analyses were performed using Graphpad Prism 9 (Graphpad Software Inc.). Data from FRET experiments were analyzed using one-way analysis of variance (ANOVA) and Tukey’s multiple-comparison posttest, while data from BRET assays were analyzed with the unpaired Student t test, with Welch’s correction. Differences between groups were considered to be significant at a *p*-value of <0.05.

## 3. Results

### 3.1. ppUL44 Residues Leucine 86/87 and Isoleucine 135 Are Crucial for Its Dimerization and Interaction with pUL54 in Cells, Respectively

We have previously shown that different ppUL44 substitution derivatives can reciprocally influence their subcellular localization, as well as that of pUL54 C-terminal domain, by CLSM analysis of GFP and DsRed2 fusions, suggesting functional interaction in a cellular context [13,23]. However, interpretation of such relocalization assays could be complicated by experimental artefacts due to DsRed2 oligomerization [41]. We therefore decided to further corroborate such findings by fluorescence energy transfer (FRET) assays. To this end, HEK293T cells were transfected with plasmids mediating the expression of CFP-UL44 or its L86A/L87A and I135A derivatives, expected to be impaired in dimerization and in binding to pUL54, respectively. Cells were also co-transfected with plasmids expressing YFP-tagged ppUL44 and the C-terminus of pUL54 bearing the ppUL44-binding domain (amino acids 1125–1242). We initially assessed the subcellular localization of each fusion protein when individually expressed (Figure 2A, left panels). As expected, both control proteins CFP and CFP-YFP localized with a diffused pattern throughout the cell. On the other hand, CFP-UL44 localized in the cell nucleus, with a punctate pattern, dependent on its ability to bind to dsDNA [26]. The dimerization defective CFP-UL44-L86/L87A still localized within the cell nucleus, due to the presence of a functional NLS, but with a diffuse pattern, consistent with its reduced DNA-binding ability [23]. CFP-UL54(1125–1242), similarly containing a functional NLS, but devoid of DNA-binding properties, localized to the cell nucleus with a diffuse pattern [13]. Co-expression of CFP- and YFP-tagged ppUL44 resulted in the two proteins colocalizing into the cell nucleus with evident accumulation in punctate structures, while co-expression of CFP- and YFP-tagged ppUL44-L86A/L87A derivatives led to co-localization within the nucleus, but with a diffuse pattern (Figure 2A, middle panels). Importantly, upon co-expression with YFP-UL44, CFP-UL54(1125–1242) relocalized to nuclear punctate structures, while its localization mainly remained diffused upon co-expression with YFP-UL44-I135A. Our FRET analysis (Figure 2B), revealed a significantly higher FRET efficiency for the CFP-YFP fusion protein (33.2 ± 0.5; *n* = 20), as compared to for the co-expression of CFP and YFP as individual proteins (8.6 ± 0.9; *n* = 24; *p* < 0.0001). On the other hand, co-expression of CFP- and YFP-tagged ppUL44 resulted in a significantly higher FRET efficiency (28.4 ± 5.2; *n* = 26) as compared to their L86A/L87A counterparts (9.3 ± 4.2; *n* = 18; *p* < 0.0001). Similarly, expression of CFP-tagged pUL54 in the presence of ppUL44 resulted in a significantly higher FRET efficiency (18.9 ± 3.8; *n* = 27) as compared to the FRET efficiency in the presence of a ppUL44 mutant bearing the I135A substitution, which prevents the interaction with pUL54 in vitro (6.3 ± 3.0; *n* = 40; *p* < 0.0001). In order to avoid artifacts due to variable expression levels among the different plasmids used for transfection, FRET data were collected from cells expressing similar levels of CFP and YFP fusion proteins (Figure 2C,D). As a result, correlation between protein expression levels and FRET efficiencies was below 0.5 for all FRET pairs (Appendix A). Clearly, these data indicate that full-length ppUL44 exists as a dimer in a cellular context, and it can interact with pUL54 in the absence of other viral proteins. Furthermore, while dimerization relies on residues identified at the dimerization interface in the previously published the crystal structure of its N-terminal domain, such as L86 and L87, its interaction with pUL54 relies on connector loop residues, such as I135.

### 3.2. BRET Assays Allow to Quantify ppUL44 Homodimerization and Holoenzyme Formation in Living Cells

Previous studies proposed the ppUL44-pUL54 interaction as a potential target for therapeutic intervention and identified SMs inhibiting HCMV replication by interfering with holoenzyme formation [27,28,29]. In vitro, the affinity of ppUL44 self-interaction has been reported to be similar to that of the ppUL44-pUL54 interaction, although using different methods and thus not being directly comparable [4,9]. We therefore decided to compare the affinity of such interactions using a quantitative BRET assay that we recently developed [31]. To this end, we generated expression vectors mediating the expression of Renilla luciferase (RLuc) fusion proteins either with full-length ppUL44 (pDESTnRLuc-UL44) or with its C-terminal 28 aa (pDESTnRLuc-UL44(405–433)), lacking the ability to self-associate, which therefore represents an ideal negative control for BRET assays. When transiently expressed in HEK293T cells, both proteins localized to the cell nucleus upon CLSM subcellular localization analysis (Figure 3A, top panels) and could be detected by Western blotting at the expected apparent molecular weight of 84 and 41 kDa, respectively (Figure 3A, bottom panels). Subsequently, HEK 293T cells were transfected to express either RLuc-UL44 or RLuc-UL44(405–433) either in the absence or in the presence of YFP-UL44, and 48 h later cells were processed for BRET analysis. Importantly, under all conditions tested, both RLuc (Figure 3B) and YFP fusions (Figure 3C) were expressed to comparable levels. When individually expressed, RLuc-UL44 generated a BRET value of 0.31 ± 0.01, similar to the 0.32 ± 0.01 BRET value calculated for Rluc-UL44(405–433) and thus corresponding to the background signal (Figure 3D). Importantly, co-expression with YFP-UL44 increased the RLuc-UL44 BRET value to 0.67 ± 0.05, resulting in a BRET ratio of 0.36 ± 0.10. On the other hand, BRET value of RLuc-UL44(405–433) was not increased by co-expression with YFP-UL44 (0.35 ± 0.01), resulting in a BRET ratio of 0.03 ± 0.00 (Figure 3E).

Subsequently, we performed BRET saturation experiments by transfecting cells with a fixed amount of BRET donor plasmid RLuc-UL44 or RLuc-UL44(405–433) in the presence of increasing amounts of BRET acceptor plasmid YFP-UL44, the BRET ratio relative to RLuc-UL44 and YFP-UL44 promptly reached saturation, whereas the BRET ratio relative to RLuc-UL44(405–433) and YFP-UL44 did not (Figure 3F). This indicates that the observed BRET signal relative to RLuc-UL44 and YFP-UL44 is specific, and not due to protein overexpression. Data fitting allowed us to calculate the B_max_ value, corresponding to the maximal BRET ratio obtainable for a BRET pair, whose intensity depends on the efficiency of the energy transfer between donor and acceptor molecules (and therefore their proximity and dipole orientation). We could also calculate the B_50_ value, corresponding to the ratio between BRET acceptor and donor sufficient to generate a BRET ratio corresponding to half of the B_max_, and indicative of the affinity of ppUL44 self-interaction in living cells (see Table 1).

### 3.3. ppUL44 Dimerizes in Cells with an Affinity Comparable to That of the ppUL44-pUL54 Interaction

Since BRET saturation experiments allowed to estimate the affinity of ppUL44 dimerization, we aimed at setting up similar assays for the ppUL44/pUL54 interaction, thus being able to compare the affinity of both processivity factor multimerization and DNA polymerase holoenzyme formation. To this end we generated an expression plasmid allowing expression of a fusion protein between RLuc and pUL54 C-terminal domain (residues 1125–1242), encompassing its NLS and the ppUL44 binding site. Transient transfection of HEK293T cells revealed nuclear localization, as assessed by CLSM, and an apparent molecular weight of about 48 kDa, as assessed by Western blotting (Figure 4A). We then used such plasmid in combination with pDESTnYFP to perform similar BRET saturation experiments for RLuc-UL54/YFP-UL44, titrating the amount of RLuc plasmids for transfection in order to ensure similar expression levels between RLuc-UL44 and RLuc-UL54 (Figure 4B). Importantly, ppUL44 appeared to dimerize with an affinity similar to that of the pUL54–ppUL44 interaction, as suggested by the very similar B_50_ values (19.0 ± 13.7; *n* = 8 and 20.0 ± 13.9; *n* = 5; see Figure 4B,H,K and Table 1).

### 3.4. Determination of the Impact of Specific Amino Acids Substitutions within ppUL44 Functional Domains on Protein Homodimerization and DNA Polymerase Holoenzyme Formation

The same approach was then used to study the effect of specific ppUL44 substitutions on homodimerization and pUL54 binding. In addition to the dimerization interface substitution derivative ppUL44-L86A/L87A and the connector loop substitution derivative ppUL44-I135A, the ppUL44 flexible gap loop substitution derivative ppUL44Δloop, impaired for DNA binding but neither for ppUL44 dimerization nor for pUL54 binding, and ppUL44ΔNLS, impaired for nuclear translocation, were also included (see Figure 1 for schematic representation). To allow proper comparison between the different ppUL44 derivatives, appropriate transfection conditions for each BRET pair were established, resulting in similar RLuc-fusion expression levels both when investigating ppUL44 dimerization (Figure 4G) and its interaction with pUL54 (Figure 4J). Importantly, substitutions at the ppUL44 dimerization interface (L86A/L87A) significantly increased the B_50_ value relative to ppUL44 dimerization (Figure 4D,H; *p* = 0.0020) but not that relative to binding to pUL54 (Figure 4D,K). On the other hand, the I135A substitution significantly increased the B_50_ value relative to binding to pUL54 (Figure 4C,K; *p* = 0.0321) but not that relative to ppUL44 dimerization (Figure 4C,H). These two observations further confirm that distinct and independent ppUL44 residues are involved in the two processes. Importantly, the Δloop (Figure 4E) and ΔNLS (Figure 4F) substitutions did not significantly affect the B_50_ relative to either interaction (Figure 4H,K), further confirming that holoenzyme formation can occur in the absence of dsDNA, and suggesting that dsDNA binding does not influence the affinity between ppUL44 and pUL54 (see Figure 4E and Table 1). However, both the L86A/L87A and the Δloop substitutions significantly decreased the B_max_ values relative to both ppUL44 dimerization and pUL54/ppUL44 interaction (Figure 4D,E,I,L) implying the possibility that DNA binding influences the conformation of both ppUL44 homodimers and of the holoenzyme complex.

### 3.5. Generation of YFP-UL44 Stable Cell Lines

Our data clearly indicate that BRET assays allow to easily monitor the interaction of biologically relevant protein–protein interactions crucial for HCMV genome replication, and they could potentially be used to screen compounds inhibiting such interactions and thus viral replication. Since previous studies implied that expression dynamics of BRET donor and acceptor proteins could influence the success of such approaches in drug discovery [39], we aimed at comparing three different expression systems for BRET pairs: (i) the most simple expression system relies on transient expression of both BRET donor and acceptor proteins, and corresponds to the approach already described in Figure 3 and Figure 4; other expression systems proposed rely either on (ii) stable expression of the BRET acceptor (YFP-fusion protein) and transient expression of the BRET donor (RLuc-fusion protein) or on (iii) stable expression of both BRET pairs. Therefore, we set out to generate YFP-UL44 stably expressing cells. To this end, we transduced HEK293A either with lentiviral particles mediating the expression of YFP-UL44 and a puromycin resistance marker or with Lentiviral particles mediating the expression of the puromycin resistance marker alone (Figure 5A, panel 1). In order to obtain cells with different levels of transgene expression, cells were transduced once, twice and three times. Positively transduced cells were selected with puromycin (Figure 5A, panel 2) and propagated to obtain the polyclonal cell lines 68 × 1, 68 × 2 and 68 × 3 (mediating expression of YFP-UL44 and the puromycin resistance marker) as well as 7 (mediating expression of the puromycin resistance marker) (Figure 5A, panel 3). Fluorometric analysis of protein expression levels and BRET assays after transfection with RLuc-UL44 revealed that cell line 68 × 3 allowed higher YFP-UL44 expression and BRET ratio, and was chosen for further experiments (Appendix A).

Cells were subsequently seeded in 96 well plates at 0.5/cells well in order to obtain monoclonal cell lines (Figure 5A, panel 4), which were further expanded. All cell lines were regularly monitored for YFP-UL44 fluorescence intensity using an inverted fluorescence microscope. Twelve cell clones expressing different levels of YFP fluorescence were selected for further analysis. Our analysis revealed that while the polyclonal cell line exhibited a mixed fluorescence intensity and grew regularly, being passaged 1:6 every week, the monoclonal cell lines exhibited more homogeneous expression levels, with higher expression linked to a marked delay in cell growth, to the point that the highest expressing cell lines (2G7, 2D11, 1A7) could not be passaged more than once (Appendix A). On the other hand, eight out of the remaining clones could be passaged indefinitely, although with different rates, with quicker growth associated to lower YFP-UL44 expression levels. These data suggest that the ability of ppUL44 to bind to cellular dsDNA can interfere with cell duplication and survival. We then selected 5 monoclonal cell lines and compared them to the polyclonal cell lines 68 × 3 and 7 in terms of (i) YFP-UL44 subcellular localization by microscopic analysis (Figure 5B); (ii) YFP-UL44 expression levels by fluorimetry (Figure 5C); and (iii) BRET assays upon transient transfection with pDESTnRLucUL44 (Figure 5D–G). Our results indicate that in all cell lines, with exception of 1G2, whereby no signal could be detected, YFP-UL44 localized to the cell nucleus as expected (Figure 5B), and all generated a detectable amount of fluorescence as assessed by fluorometric analysis (Figure 5C). Most importantly, BRET ratios for all monoclonal cell lines (range 0.18–0.24) was higher than that for calculated for the polyclonal cell line 68 × 3 (Figure 5F,G). Cell line 1B2, which exhibited the highest BRET ratio was selected for further analysis. Overall, our results indicate that stable expression of YFP-UL44 at high levels can interfere with cell growth, and cells stably expressing YFP-UL44 at specific levels can be cultured for a long time and generate a detectable BRET signal upon transient expression of RLuc-UL44.

### 3.6. Generation of YFP-UL44/RLuc-UL44 Stable Cell Lines

Monoclonal cells 1B2 were transfected with either pDESTnRLuc-UL44 or pDESTnRLuc-UL44(405–433) as a control, selected in the presence of G418, and used to generate monoclonal cell lines as described above (Figure 6A). Several cell lines were propagated and analyzed for YFP (Figure 6B) and RLuc (Figure 6C) expression, as well as for BRET values (Figure 6D) and BRET ratio (Figure 6E) upon addition of coelenterazine. Importantly, for 6 of the 12 YFP-UL44/RLuc-UL44 cell lines, the selected a BRET ratio higher than zero could be measured (Figure 6E). The highest of such BRET ratios (monoclonal cell line G10F; 0.17) was extremely close to half of the B_max_ value measured for the RLuc-UL44/YFP-UL44 BRET pair in transient transfection assays (Figure 4) and to the BRET ratio obtained after transient transfection of 1B2 with RLuc-UL44 (Figure 5), and was therefore used for subsequent experiments.

### 3.7. Transient Expression of RLuc-UL44 Allows to Monitor Interference with RLuc-UL44/YFP-UL44 and RLuc-UL54/YFP-UL44 Interactions upon Overexpression of FLAG-UL44

We then compared the effect of FLAG-UL44, and of the non-interacting FLAG-UL42 fusion proteins on the BRET signal originated by expression of RLuc-UL44 and YFP-UL44 using the three different systems mentioned above (Figure 7A). Therefore HEK293T, 1B2 and G10F cells were transiently transfected with plasmids encoding for either RLuc-UL44 and YFP-UL44, RLuc-UL44 or pCDNA3, respectively, in the presence of increasing concentrations of plasmids pDESTnFLAG-UL44 or pDESTnFLAG-UL42 (Figure 7A). Interestingly, transfection with pDESTnFLAG-UL44, but not pDESTnFLAG-UL42, resulted in a concentration-dependent decrease in the BRET signal when RLuc-UL44 was transiently expressed and YFP-UL44 was either transiently (system 1, Figure 7B, left panel) or stably expressed (system 2, Figure 7B, middle panel). BRET inhibition did not depend on the reduction of YFP-UL44 expression levels (Figure 7C, compare data for FLAG-UL44 and FLAG-UL42). However, when both RLuc-UL44 and YFP-UL44 were stably expressed before transfection with pDESTnFLAG-UL44 (system 3, Figure 7B, right panel), no inhibition of BRET signal was detected. We therefore applied a transient expression approach to perturb the BRET signal relative to the ppUL44/pUL54 interaction. To this end, we transiently transfected HEK293T cells with fixed amounts of pDESTnRLuc-UL54 and pDESTnYFP-UL44 and increasing concentrations of pDESTnFLAG-UL44 or pDESTnFLAG-UL42 as a control (Figure 7C). Our data revealed that overexpression of FLAG-UL44 could also inhibit, although less efficiently, the BRET signal generated from the RLuc-UL54/YFP-UL44. Indeed, transfection with up to 900 ng of pDESTnFLAG resulted in about 40% inhibition, as compared to >90% inhibition relative to RLuc-UL44/YFP-UL44 (Figure 7D, left panel). In this case, no significant inhibition was observed upon overexpression of the unrelated FLAG-UL42 fusion protein (Figure 7D, middle panel), and inhibition was not dependent on reduction of the YFP-UL44 expression levels (Figure 7D, right panel).

## 4. Discussion

This is the first quantitative study of HCMV DNA polymerase holoenzyme complex formation in living cells, which confirms and further extends previous findings from our and other groups [4,9,12,13,14,23,26,42]. Our data indicate that in cells, the affinity of the ppUL44 dimerization is similar to that of the ppUL44/pUL54 interaction (see Table 1, Figure 4). Such finding is in accordance with data obtained with bacterially purified UL44Δ290 which homodimerizes with a Kd of~250 nM in gel filtration assays [4] and binds to the C-terminal 20 aa of pUL54 with a Kd of~700 nM, as assessed by isothermal titration calorimetry [9]. Furthermore, our data clearly confirmed that both processes can occur both in the nucleus [13,23]. Indeed, a NLS defective substitution derivative of ppUL44, whose localization is strictly cytosolic [18], could both self-interact and bind to pUL54 with very similar B_50_ values as compared to the wild-type protein (see Table 1, Figure 4F). The fact that a nuclear localization defective ppUL44 can both homodimerize and bind to pUL54 as efficiently as the wild-type protein also suggests that dsDNA binding does not influence ppUL44 dimerization nor pUL54 biding. The latter hypothesis is also supported by the ability of the ppUL44Δloop substitution derivative, which is severely impaired for dsDNA binding both in vitro and in cells [24,26], to both dimerize and bind to pUL54 with similar affinity as compared to the ppUL44 wild-type. However, dsDNA binding of ppUL44 could result in conformational changes, both of ppUL44 homodimers and of the DNA polymerase holoenzyme. This is suggested by the significantly lower B_max_ values calculated for all ppUL44 derivatives endowed with impaired DNA binding such as the ΔNLS and the Δloop derivatives both for dimerization and pUL54 binding. Conformational changes have been described for a number of proteins upon DNA binding [43], and although a crystal structure for ppUL44 complexed with dsDNA is not available yet, it is tempting to speculate that its dimeric “open” nature would allow more flexibility as compared to the trimeric “closed” PCNA upon DNA binding [43]. Indeed, binding to pUL54 C-terminal residues has been shown to remarkably affect the ppUL44 homodimer conformation, with effects on its dsDNA binding abilities [14].

Besides the effect of dsDNA binding on ppUL44 on its conformation, it appears clear that specific ppUL44 residues are involved in its ability to interact with different partners, such as in the case of PCNA [44], and quantitative BRET assays could help further characterize such important protein–protein interactions. Finally, the systems developed here for monitoring both ppUL44 dimerization and DNA polymerase holoenzyme formation have potential implications in drug discovery. In this respect, our finding that transient expression of a BRET donor is required for inhibition of energy transfer upon overexpression of a competitor (Figure 7A–C) is consistent with the idea that preventing interaction of the BRET donor with its BRET acceptor partner is more feasible than disrupting an already formed protein complex [45].

## Figures and Tables

**Figure 1 microorganisms-09-00928-f001:**
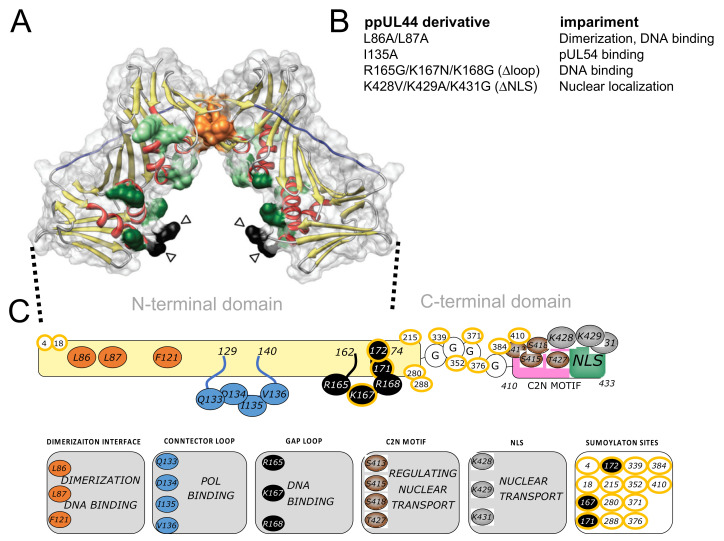
ppUL44 structure, functional domains and key residues. (**A**) The crystal structure of pUL44 (1–290; pdb code: 1T6L) is shown as ribbons with transparent surface (red, α-helices; yellow, β-sheets). Key residues involved in dimerization (orange), DNA binding (green), as well as those located within the connector (blue) or gap (black) loops are shown in color. Empty arrowheads indicate residues that are not visible in the electron density map. Copyright ^©^ American Society for Microbiology, Journal of Virology, 83, 9567–9576, 2009. (**B**) List of ppUL44 substitution derivative tests in this study, along with their defects as compared to the wild-type protein. (**C**) Schematic representation of ppUL44 functional domains with highlighted residues involved in specific functions. Residues involved in dimerization are in orange, the connector loop is in blue, the gap loop is in black. The C2N motif is in pink, the nuclear localization signal (NLS) in cyan with key residues in brown and grey, respectively. Sumoylated residues have a yellow outline.

**Figure 2 microorganisms-09-00928-f002:**
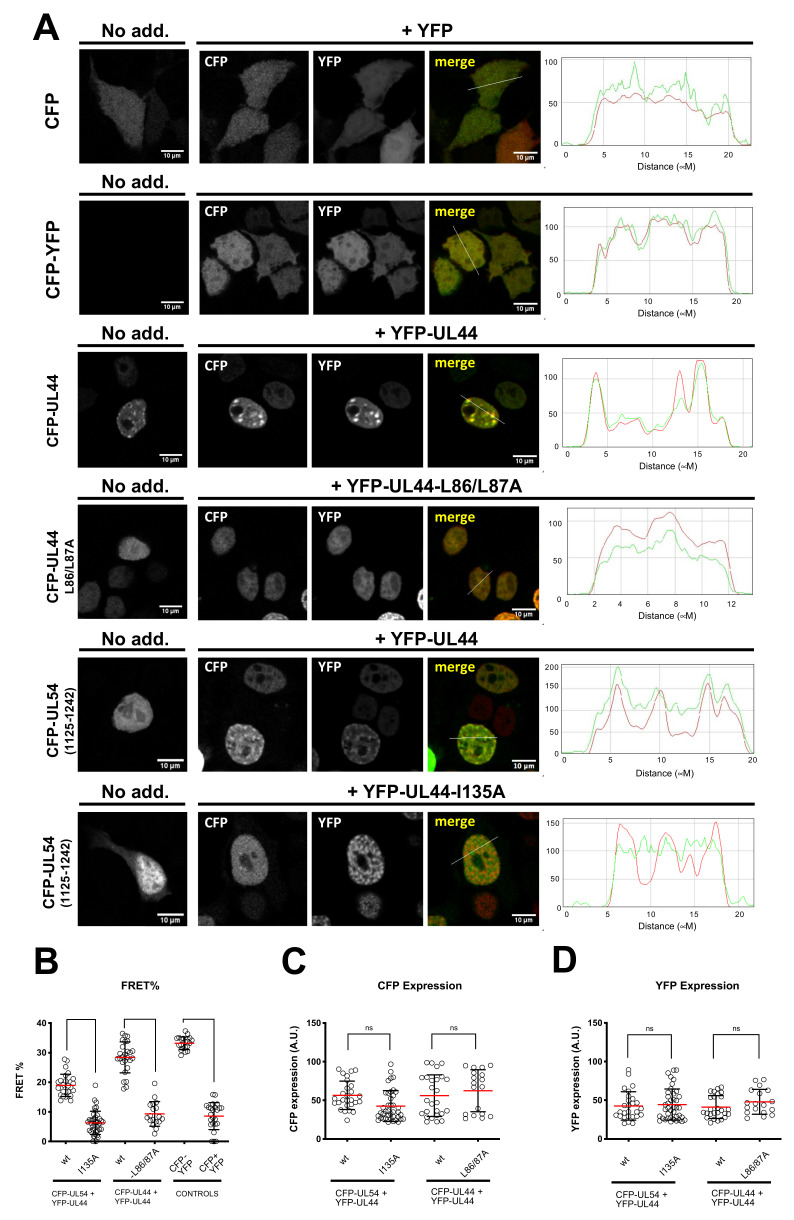
Specific residues are involved in ppUL44 dimerization in cells. HEK293T cells were seeded on glass coverslips and transfected with the indicated expression plasmids; 48 h p.t., cells were fixed and processed for IF, before being analyzed by confocal laser scanning microscopy (CLSM) to investigate the subcellular localization of the indicated fusion proteins. (**A**) The subcellular localization of CFP fusions as expressed alone (left panels) or in the presence of YFP fusions (middle panels) is shown, together with the correspondent RGB profiles (right panels). Cell micrographs such as those shown in (**A**), were subjected to FRET acceptor photobleaching to calculate fluorescence resonant energy transfer (FRET) efficiency (**B**), as well as the CFP (**C**), and YFP (**D**) expression level of individual cells relative to each indicated FRET pair, as described in the Materials and Methods section. Data shown are the mean + standard deviation of the mean relative to at least 20 cells from two independent experiments, along with the *p*-value relative to the indicated groups of FRET pairs, calculated using the ordinary one-way ANOVA with Tukey correction: **** = *p* ≤ 0.0001.

**Figure 3 microorganisms-09-00928-f003:**
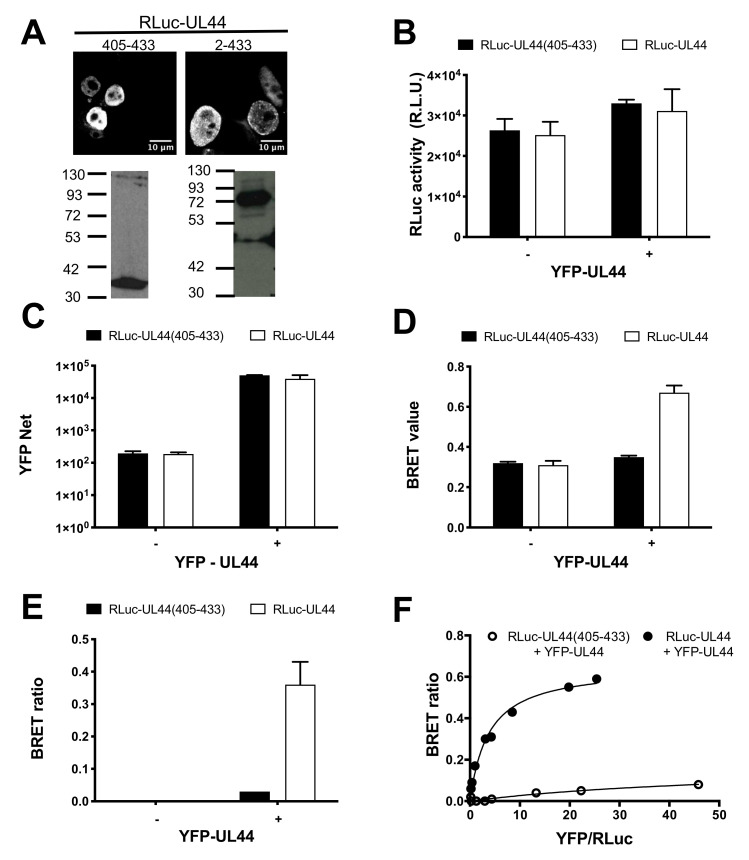
Bioluminescence resonant energy transfer (BRET) analysis of UL44 self-interaction in living cells. HEK293T cells were transfected to transiently express either RLuc-UL44(2–433) or RLuc-UL44(405–433), either in the absence or in the presence of YFP-UL44; 48 h later cells were processed for immunofluorescence and Western blotting (**A**) or BRET measurements (**B**–**F**) as described in the Materials and Methods section. (**A**) Subcellular localization and molecular weight of Renilla luciferase (RLuc)-fusion proteins was assessed using a mouse monoclonal antibody against RLuc. For BRET assays, RLuc activity (**B**) and YFP fluorescence (**C**) were measured, and the BRET value (**D**) and BRET ratio (**E**) were calculated. Representative data from two independent experiments performed in triplicate are shown. (**F**). The BRET ratio relative to indicated BRET pairs is plotted against the ratio between YFPnet and RLuc emission as described in the Materials and Methods section. Representative data from two independent experiments performed in triplicate are shown.

**Figure 4 microorganisms-09-00928-f004:**
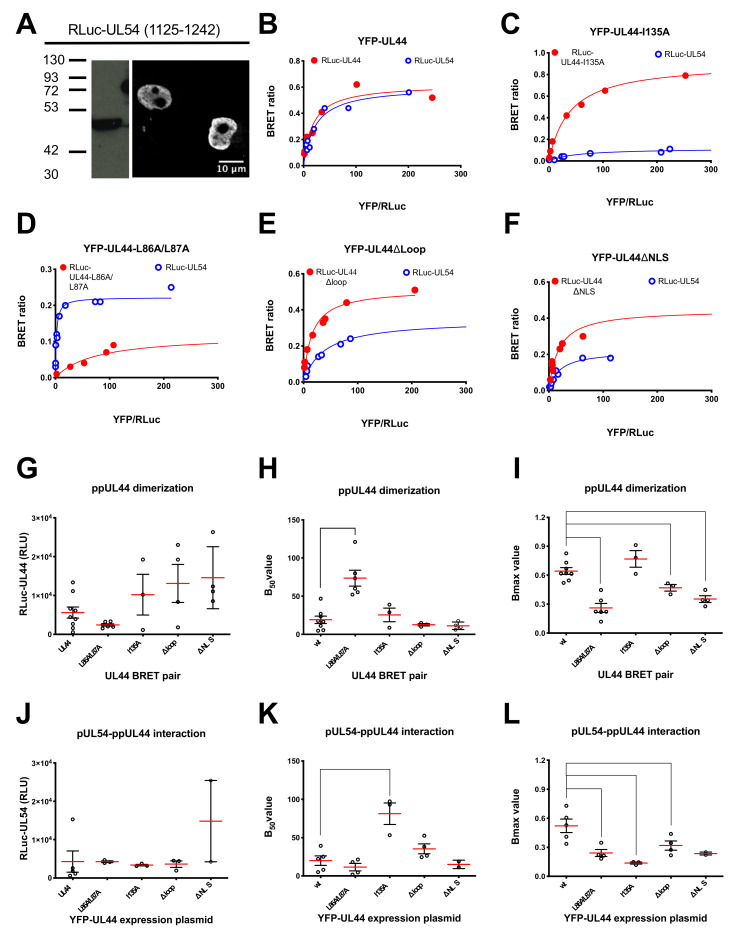
Assessment of the effect of substitutions within ppUL44 individual domains differently affect protein dimerization and holoenzyme formation in living cells. HEK293T cells were transfected to transiently express the indicated RLuc-UL44 (red) or RLuc-UL54 (blue) fusion proteins either alone or in the presence of increasing amounts of plasmids encoding for YFP-tagged ppUL44 proteins indicated in Figure 1B (range 0–500 ng); 48 h later, cells were processed for immunofluorescence and Western blotting (**A**) or BRET measurements as described in the Materials and Methods section (**B**–**L**). Subcellular localization and molecular weight of RLuc-UL54 was assessed using a mouse monoclonal antibody against RLuc (**A**). The BRET ratio relative to the indicated BRET pair was plotted against the YFPNet/RLuc ratio to generate BRET saturation curves (**B**–**F**). Representative data from at least three independent experiments are shown. RLuc activity (**G**,**J**), B_50_ (**H**,**K**) and BRET ratio (**I**,**L**) values relative to the indicated YFP-UL44 substitution derivatives in combination with either the corresponding RLuc-UL44 derivatives (**G**–**I**) or RLuc-UL54 (**J**–**L**) are shown as individual measurements (empty circles) and means (red horizontal lines) ± standard error of the mean (black vertical lines), relative to at least three independent experiments, along with the *p*-value as compared to wild-type UL44, calculated using the unpaired t-student test with Welch’s correction: **** = *p* ≤ 0.0001; ** = *p* < 0.005; * = *p* ≤ 0.05.

**Figure 5 microorganisms-09-00928-f005:**
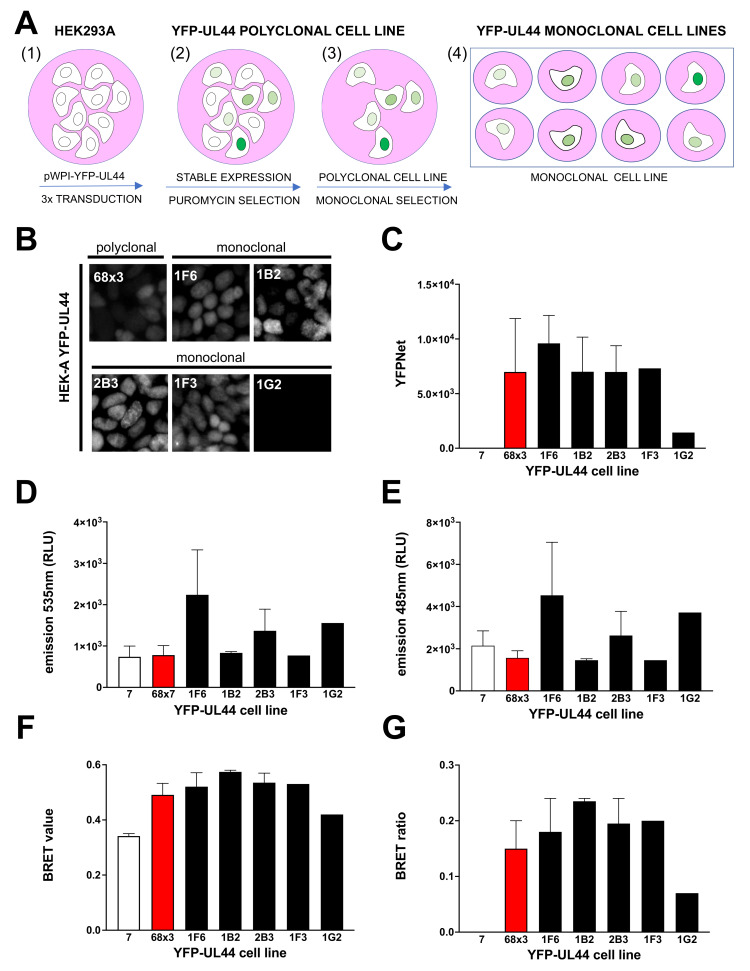
Establishment of polyclonal and monoclonal cell lines stably expressing different levels of YFP-UL44 for BRET assays. (**A**) HEK293A cells were seeded in 6-well plates and, starting 24 h later, were sequentially transduced 1, 2 or 3 times with Lentiviral particles containing either pWPI-puro or pPWI-YFP-UL44-puro (1). Transduced cells were selected with puromycin (2) and expanded to generate stable cell lines (3). The latter were diluted to single cells in 96-well plates and expanded to generate monoclonal cell lines (4). YFP-UL44 subcellular localization was microscopically evaluated (**B**), whereas its expression levels were quantified by fluorimetry (**C**). Upon transfection with 25 ng of pDESTnRLuc-UL44 and addition of coelenterazine, the indicated cell lines were evaluated for 535 (**D**) and 485 (**E**) nm emission, allowing to calculate their respective BRET values (**F**) and BRET ratios (**G**).

**Figure 6 microorganisms-09-00928-f006:**
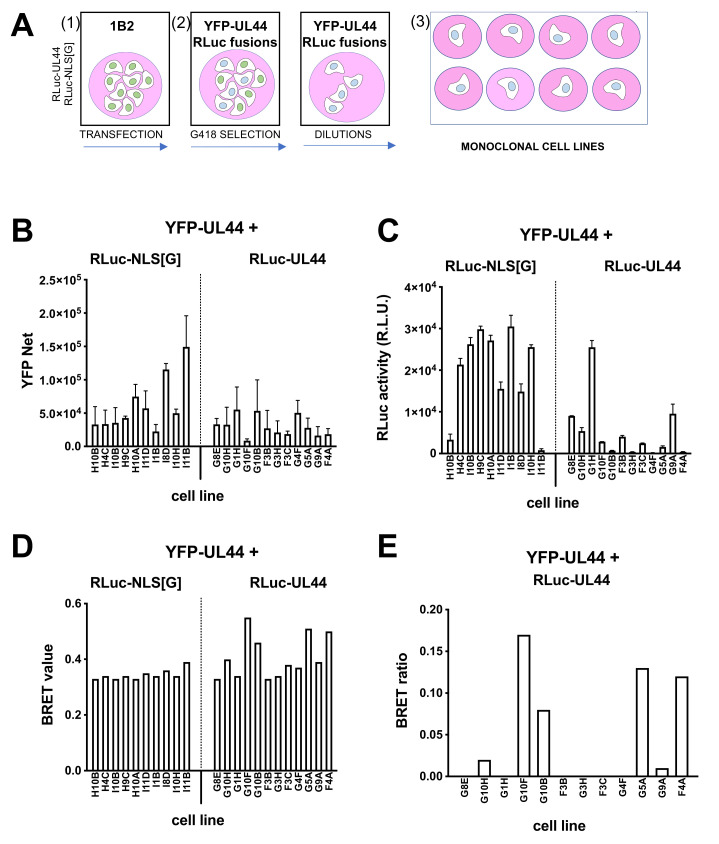
Establishment of and monoclonal cell lines stably expressing YFP-UL44 and different levels of RLuc-UL44 for BRET assays. (**A**) 1B2 monoclonal cell line was transfected (1) with either pDESTnRLuc-UL44 or pDESTnRLuc-NLS[G], and starting 48 h later, selected with G418 (2) and seeded at 0.5 cells/well in 96-well plates to generate monoclonal cell lines stably expressing both RLuc- and YFP-UL44 (3). YFP expression was evaluated by fluorescent emission (**B**), and, upon addition of coelenterazine, all cell lines were tested in BRET assays to quantify RLuc expression (**C**), allowing to calculate their respective BRET values (**D**) and BRET ratios (**E**).

**Figure 7 microorganisms-09-00928-f007:**
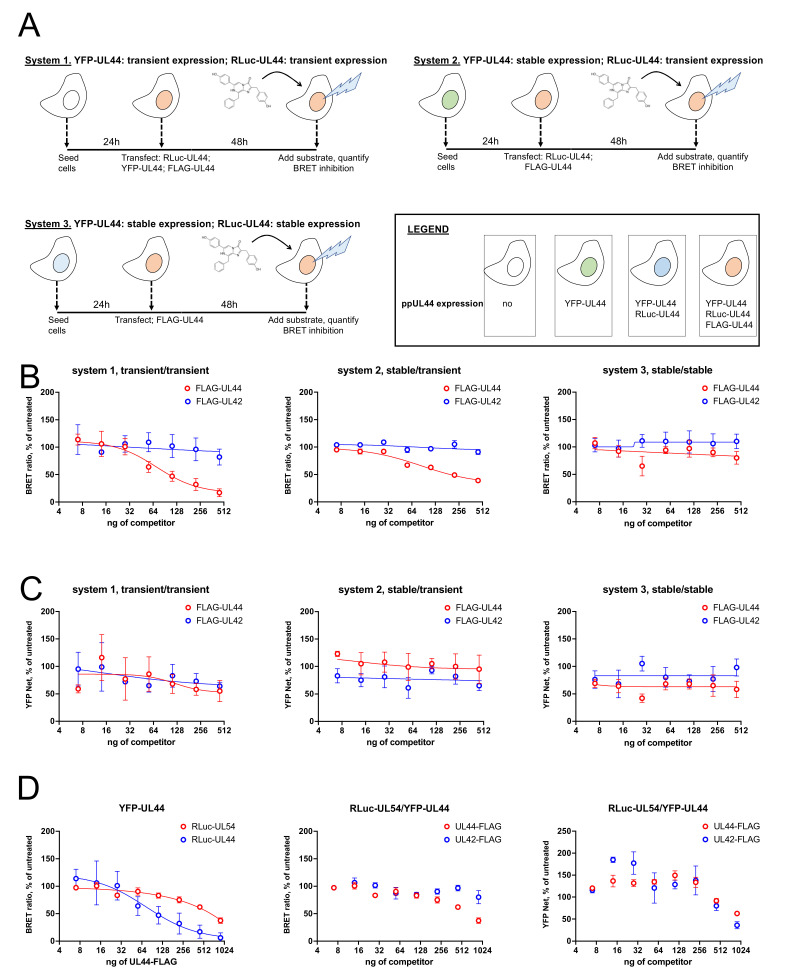
Stable expression of RLuc-UL44 and RLuc-UL54 allows to detect a concentration-dependent decrease in BRET ratio upon overexpression with FLAG-UL44. (**A**). The three different expression systems compared for detection of inhibition of BRET signal upon overexpression of a known competitor are shown. After carefully selecting condition to generate similar BRET rations, cells were transfected with increasing amounts of either pDESTnFLAG-UL44 as a competitor or pDESTnFLAG-UL42 (range 0–500 ng) as a control. At 48 h post transfection, cells were subjected to fluorometric and luminescence measurements to detect BRET ratios (**B**) and YFP expression (**C**) relative to each condition. Data shown are the mean + relative to at least three independent experiments. HEK293T were also transiently transfected with appropriate amounts of RLuc-UL54 and YFP-UL44 expression plasmids in the presence of increasing concentrations of either pDESTnFLAG-UL44 as a competitor, or pDESTnFLAG-UL42 as a control (range 0–1000 ng). At 48 h post transfection, cells were subjected to fluorometric and luminescence measurements to calculate BRET ratios (**D**; left and middle panels) and YFP expression (**D**; right panels) relative to each condition. A RLuc-UL54/YFP-UL44 inhibition of BRET ratio upon FLAG-UL44 expression is shown as compared to that obtained for RLuc-UL44/YFP-UL44 (**D**; left panel) or for FLAG-UL42 overexpression (**D**; middle panels). Data shown are the mean + relative to at least three independent experiments.

**Table 1 microorganisms-09-00928-t001:** BRET saturation curves such as those shown in Figure 4B–F were analyzed to calculate the B_max_ and B_50_ values relative to the indicated BRET pairs. Data shown are the mean and standard deviations of the mean. The numbers between brackets indicate the number of independent experiments relative to each BRET pair. ^1^ Substitutions that impair ppUL44 dimerization and reduce DNA binding in vitro [4]. ^2^ Substitutions that interfere with ppUL44 DNA binding in cells [26]. ^3^ Substitutions that prevent transcomplementation of oriLyt-dependent DNA replication [19,23,26]. ^4^ Substitution that prevents ppUL44 binding to pUL54 in vitro [9]. ^5^ Substitution that prevents nuclear import [18].

UL44 Derivative	RLuc-UL44 + YFP-UL44		RLuc-UL54 + YFP-UL44	
	B_max_	B_50_	B_max_	B_50_
Wild-type	0.64 ± 0.09 (8)	19.0 ± 13.7 (8)	0.52 ± 19.56 (5)	20.0 ± 13.9 (5)
L86A/L87A ^1,2,3^	0.26 ± 0.11 (6)	73.7 ± 25.7 (6)	0.24 ± 0.07 (4)	11.5 ± 9.8 (4)
I135A ^3,4^	0.77 ± 0.15 (3)	25.5 ± 15.4 (3)	0.14 ± 0.02 (3)	81.2 ± 24.3 (3)
Δloop ^2,3^	0.47 ± 0.06 (3)	12.6 ± 2.6 (3)	0.32 ± 0.10 (4)	34.4 ± 13.0 (4)
ΔNLS ^3,5^	0.35 ± 0.07 (4)	11.2 ± 4.9(4)	0.24 ± 0.02 (2)	15.1 ± 5.4 (2)

## Data Availability

All data is available from the corresponding author upon request.

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
