# Peer review of "Live-Cell Analysis of Human Cytomegalovirus DNA Polymerase Holoenzyme Assembly by Resonance Energy Transfer Methods"

_microorganisms, 2021, doi:10.3390/microorganisms9050928_

Round 1
Reviewer 1 Report
The paper by Di Antonio and coworkers is focused on the analysis of HCMV DNA polymerase holoenzyme assembly. This paper is well written and shed new light on ppUL44 and pUL54 interaction showing new methodological approaches on this major item of HCMV biology.
Minor concerns:
- Several standard deviations are indicated without the negative value (i.e. page 8 line 267 33.2+0.5 has to be corrected as 33.2+5).
- Please, indicate the standard deviation bar in Figure 3 panel F and in the other similar panels exhibited in subsequent figures (Figures 4 and 6).
- Please, add the p value in the text and not only in the figures.
- The Authors indicate the use of Student t test in Figure 2 but this choice is not commented in the Material and Methods section.
In my opinion, this is an interesting paper that may be accepted for publication after the corrections of these minor concerns.
Author Response
The paper by Di Antonio and coworkers is focused on the analysis of HCMV DNA polymerase holoenzyme assembly. This paper is well written and shed new light on ppUL44 and pUL54 interaction showing new methodological approaches on this major item of HCMV biology.
We thank the Reviewer for his/her very positive feedback of our work
Minor concerns:
- Several standard deviations are indicated without the negative value (i.e. page 8 line 267 33.2+0.5 has to be corrected as 33.2+5).
We thank the Reviewer for pointing out such issue, which has now been fixed.
- Please, indicate the standard deviation bar in Figure 3 panel F and in the other similar panels exhibited in subsequent figures (Figures 4 and 6).
We must highlight than it is not always possible to show an error bar. Indeed, error bars are present when possible. Indeed error bars cannot be shown for representative BRET saturation experiments such as those shown in Figure 3 panel F and similar, whereby representative curves are shown. In such cases, it is not meaningful to show SD for BRET values, since the NetYFP/Rluc values (x-axis values) are slightly different in each experiment. This would be possible if we plotted BRET values against plasmid concentrations (which are the same for every experimental replicate), instead for actual YFP Fluorescence and Renilla bioluminescence ratios (which slightly vary each time and experiment is performed, but are much more meaningful than ratios of plasmids amounts used for transfection).
Please, add the p value in the text and not only in the figures.
We apologize for the inconvenience. P values relative to data from Figure 3 and 4 have been added to the text. We thank the Reviewer for pointing out such issue.
- The Authors indicate the use of Student t test in Figure 2 but this choice is not commented in the Material and Methods section.
We apologize for the lack of appropriate explanation of the statistical analysis used. Importantly, the reviewer is correct. For FRET experiments whereby results relative to multiple different FRET pairs were compared, a one way ANOVA with Tukey correction was used, while unpaired t student was used with Welch’s correction was used to analyze data from Figure 4, whereby the effect of specific mutations are compared to the wild-type. The Material and Methods section was modified to accommodate an appropriate paragraph according to the Reviewer’s suggestions, to read “Statistical analysis. Statistical analyses were performed using Graphpad Prism 9 (Graphpad Software). Data from FRET experiments were analyzed using one-way analysis of variance (ANOVA) and Tukey's multiple-comparison posttest, while data from BRET assays were analyzed with the unpaired t-student test, with Welch’s correction. Differences between groups were considered to be significant at a P value of <0.05. “
In my opinion, this is an interesting paper that may be accepted for publication after the corrections of these minor concerns.
We are thankful to the Reviewer for his/her positive response and very pertinent comments, which greatly contributed to improving our manuscript.
Reviewer 2 Report
Di Antonio et al studied the HCMV DNA polymerase holoenzyme complex formation by resonance energy transfer methods (FRET and BRET) in live cells. HMCV DNA polymerase holoenzyme has two essential proteins, a dimeric processivity factor, ppUL44 and a catalytic subunit, pUL54 which are all required for the viral replication. The authors measured and quantified the molecular interactions between ppUL44 and pUL54 proteins as well as self -association (dimerization) of ppUL44. For this purpose, they used various ppUL44 mutants and measured the interactions either in transiently or stably ppUL44 expressing cells by taking advantage of fluorescent proteins such CPF, YFP as well as renilla luciferase. In addition, the investigators studied ppUL44 dimerization by overexpressing FLAG-ppUL44 as a competitor. The authors demonstrated that leucine residues located at 87 and 88 positions of ppUL44 are responsible for dimerization while isoleucine residue at 135 position plays a role in DNA binding. There seem be an interplay between dimerization of ppUL44 and DNA binding properties of pUL54.
The results are interesting and contributes to the understanding of the HCMV viral DNA replication. However, the text has several misspelling and typos. In addition, the text is very long and hard to read. The main text is needed to be shorten considerably. It is not acceptable for publication in its current form.
Please correct the following misspelling, typos and define the abbreviations.
Line 2. Change resonant to “resonance”.
Line 22. Change fluorescent to “fluorescence".
Line 23. Change bioluminescent resonant to ‘bioluminescence resonance”.
Line 23 …such processes.? Please define “processes”? Protein-protein interactions?
Line 87. Change fluorescent and bioluminescent to “fluorescence and bioluminescence”.
Line 157. Define CLSM. “Confocal laser scanning microscopy:”
Line 183. Fluorescent signals …. YFP fluorescent emission… should be corrected as Fluorescence signals …YFP fluorescence emission…
Figure 1B should be a separate table.
In Figure 1B, F121 is indicated that it is involved in dimerization and DNA binding. However, there is no mention of a UL44 plasmid carrying F135A mutation in the method section and, also no data was presented regarding the F121A mutant in the results.
In Figure 1B K426V/K429A/K431? What amino acid substitution was done for K431?
K426V/K429A/K431 should be defined as ΔNLS in parenthesis.
R165G/K167N/K168G should be defined s Δloop in parenthesis.
Line 242. In the subtitle instead of saying “Specific residues, mention Leucine 86, leucine 87 and isoleucine 135 residues are crucial for dimer…
Line 289. Change bioluminescent resonant to "bioluminescence resonance"…
Line 321. Change UL44 “self-interacts” to UL44 “dimerizes”….
Line 382. Change fluorescent microscope to “fluorescence” microscope…
Move Table I right after Figure 3.
Author Response
Di Antonio et al studied the HCMV DNA polymerase holoenzyme complex formation by resonance energy transfer methods (FRET and BRET) in live cells. HMCV DNA polymerase holoenzyme has two essential proteins, a dimeric processivity factor, ppUL44 and a catalytic subunit, pUL54 which are all required for the viral replication. The authors measured and quantified the molecular interactions between ppUL44 and pUL54 proteins as well as self -association (dimerization) of ppUL44. For this purpose, they used various ppUL44 mutants and measured the interactions either in transiently or stably ppUL44 expressing cells by taking advantage of fluorescent proteins such CPF, YFP as well as renilla luciferase. In addition, the investigators studied ppUL44 dimerization by overexpressing FLAG-ppUL44 as a competitor. The authors demonstrated that leucine residues located at 87 and 88 positions of ppUL44 are responsible for dimerization while isoleucine residue at 135 position plays a role in DNA binding. There seem be an interplay between dimerization of ppUL44 and DNA binding properties of pUL54.
The results are interesting and contributes to the understanding of the HCMV viral DNA replication. However, the text has several misspelling and typos. In addition, the text is very long and hard to read. The main text is needed to be shorten considerably. It is not acceptable for publication in its current form.
We thank the Reviewer for his/her very helpful comments and positive evaluation of the quality or our work. As detailed below, we sincerely apologize for the unusual amount of typos and misspellings scattered thorough the manuscript, which have all been corrected. We also acknowledge that our manuscript might feel too heavy for the readers, and have shortened as much as possible. Accordingly the introduction section has been shortened from 964 to 862 words, and the Discussion from 928 to 488 words. We hope that the Reviewer will find our writing more concise in the current version.
Please correct the following misspelling, typos and define the abbreviations.
Line 2. Change resonant to “resonance”.
We apologize for such typo. We are not native English speakers, and despite all our efforts to always carefully craft our manuscripts before peer review, we acknowledge that the outcome does not always match the standards required for publication. We corrected all such kind of typos accordingly
Line 22. Change fluorescent to “fluorescence".
The text has been modified accordingly. We thank the Reviewer for his/her patience.
Line 23. Change bioluminescent resonant to ‘bioluminescence resonance”.
See above.
Line 23 …such processes.? Please define “processes”? Protein-protein interactions?
Yes, the text has been modified as suggested by the Reviewer.
Line 87. Change fluorescent and bioluminescent to “fluorescence and bioluminescence”.
Done.
Line 157. Define CLSM. “Confocal laser scanning microscopy:”
We apologize for not having defined such an important abbreviation. The text has been modified accordingly
Line 183. Fluorescent signals …. YFP fluorescent emission… should be corrected as Fluorescence signals …YFP fluorescence emission…
We have modified all sentences as suggested by the reviewer. We apologize for misusing fluorescent and bioluminescent in place of fluorescence and bioluminescence throughout the manuscript.
Figure 1B should be a separate table.
We respectfully disagree with the Reviewer and we strongly feel that the information depicted in Figure 1B is most useful beside the panels shown in Figure 1A and 1C rather than in a separate table.
In Figure 1B, F121 is indicated that it is involved in dimerization and DNA binding. However, there is no mention of a UL44 plasmid carrying F135A mutation in the method section and, also no data was presented regarding the F121A mutant in the results.
The Reviewer is correct, and we acknowledge our mistake. It is true that the substitution F121 can decrease dimerization and DNA binding of ppUL44, however we did not perform experiments with such derivative, which has been accordingly removed from Figure 1B. We thank the Reviewer for pointing out such issue.
In Figure 1B K426V/K429A/K431? What amino acid substitution was done for K431?
Again, we apologize. The substitution inserted (K431G) has been mentioned in the revised version.
K426V/K429A/K431 should be defined as ΔNLS in parenthesis.
We have modified the Figure as suggested. We thank the Reviewer for his/her suggestion.
R165G/K167N/K168G should be defined as Δloop in parenthesis.
We have modified the Figure as suggested. We thank the Reviewer for his/her suggestion.
Line 242. In the subtitle instead of saying “Specific residues, mention Leucine 86, leucine 87 and isoleucine 135 residues are crucial for dimer…
In order to avoid any confusion but simultaneously take on board the Reviewer’s suggestion the subtitle has been changed to “ppUL44residues Leucine 86/87 and Isoleucine 135 are crucial for its dimerization and interaction with pUL54 in cells, respectively”. Indeed L86 and 87 are important for dimerization but not holoenzyme formation whereas I135 is important for holoenzyme formation but not for ppUL44 dimerization.
Line 289. Change bioluminescent resonant to "bioluminescence resonance"…
The mistake has been corrected. We thank the Reviewer for his/her suggestion.
Line 321. Change UL44 “self-interacts” to UL44 “dimerizes”….
We have modified the text as suggested.
Line 382. Change fluorescent microscope to “fluorescence” microscope…
We thank the reviewer for pointing out this issue, the text has been modified accordingly.
Move Table I right after Figure 3.
We hope that such shift can be done during proofreading. It is very difficult to move tables, figures and legends without compromising the manuscript layout.